# Expanding the Mesozoic Record of Early Brachyceran Fly Larvae, including New Larval Forms with Chimera-Type Morphologies

**DOI:** 10.3390/insects15040270

**Published:** 2024-04-13

**Authors:** André P. Amaral, Joachim T. Haug, Carolin Haug, Simon Linhart, Patrick Müller, Jörg U. Hammel, Viktor Baranov

**Affiliations:** 1Biocenter, Ludwig-Maximilians-Universität München, Großhaderner Str. 2, 82152 Planegg-Martinsried, Germany; joachim.haug@palaeo-evo-devo.info (J.T.H.); carolin.haug@palaeo-evo-devo.info (C.H.); s.linhart@campus.lmu.de (S.L.); 2GeoBio-Center at LMU, Richard-Wagner-Str. 10, 80333 München, Germany; 3Independent Researcher, 66482 Zweibrücken, Germany; pat14789@web.de; 4Institute of Materials Physics, Helmholtz-Zentrum Hereon, Max-Planck-Str. 1, 21502 Geesthacht, Germany; joerg.hammel@hereon.de; 5Estación Biológica de Doñana-CSIC, Avd. Americo Vespucio 26, 41092 Sevilla, Spain; viktor.baranov@ebd.csic.es

**Keywords:** amber, Cretaceous, Eocene, Jurassic, morphology

## Abstract

**Simple Summary:**

The majority of dipterans (flies, mosquitoes, and their allies) spend most of their lifetimes as immatures (larvae and pupae). Yet, immature dipterans are seldom reported in the fossil record. Studying this life stage can provide crucial insights into the evolution of these insects, how distinct evolutionary lineages relate to each other, and their ecological interactions during the significant changes of the Mesozoic Era (about 252–66 Ma). In this work, based on a range of imaging techniques, we describe and discuss several fossil dipteran larvae from amber and compression fossils, exhibiting forms not known in the modern fauna. These specimens possess structures similar to those found in present day larvae of awl-flies and water snipe-flies. Additionally, we report new specimens of stratiomyomorphan larvae, which appear to have dominated over other terrestrial dipteran immatures, suggesting a potentially larger ecological role in the past than in present times. Finally, we describe two additional specimens of a highly distinctive dipteran, known only as a larva, *Qiyia jurassica* Chen et al., 2014. Our findings suggest differences in the ecology of some fly groups compared to their modern relatives. They also challenge some postulated evolutionary relationships within these lineages.

**Abstract:**

Diptera are one of the four megadiverse groups of holometabolan insects. Flies perform numerous ecological functions, especially in their larval stages. We can assume that this was already the case in the past; however, fly larvae remain rare in most deposits. Here we report new dipteran larvae preserved in Cretaceous (about 99 Ma) Kachin amber from Myanmar and, even older, Jurassic (about 165 Ma) compression fossils from China. Through light microscopy and micro-CT scanning we explore their peculiar morphology and discuss their possible phylogenetic affinities. Several larvae seem to represent the lineage of Stratiomyomorpha. A few others present characters unique to Xylophagidae (awl-flies), as well as to Athericidae (water sniper-flies), resulting in a chimeric morphology. Understanding the exact relationships of most of these specimens with a particular lineage remains challenging, since they differ considerably from any other known dipteran larvae and present some unique traits. Additionally, we report new specimens of *Qiyia jurassica* Chen et al., 2014, supposedly parasitic larvae, most likely representatives of Athericidae. These new findings offer valuable insights into the evolution of the early diversification of the brachyceran flies and underscore the importance of immature stages in understanding the evolutionary history and ecology of flies.

## 1. Introduction

The diversity in continental ecosystem is dominated by four megadiverse insect lineages within the group Holometabola [1,2]: the groups Coleoptera (beetles), Hymenoptera (wasps, bees, and ants), Diptera (flies, mosquitos, and midges), and Lepidoptera (moths and butterflies). The enormous success of these groups has often been attributed to the strongly expressed differentiation between larvae (challenges of the term discussed in [3]) and adults concerning morphology, behaviour, and ecology [4,5].

Understanding the peculiarities of each life stage is therefore key to uncovering the ecological network of these ecologically important animals. It has been widely recognized that we lack information about the immature stages of most extant holometabolans, with a large part of the species only known as adults [6,7,8]. Going back to the fossil record, this problem worsens considerably given that we have even fewer tools to associate the different life stages. This way, the paleontological literature is swamped with new species descriptions based on a single sex and life stage, with little effort towards reconstructing palaeoenvironments, palaeoecologies, and the evolution of the groups (there are some exceptions for pre-Pleistocene fossils in [9,10,11,12,13,14,15,16]. It is also important to note that dipteran larvae are extremely important in the reconstruction of palaeoenvironments for Pleistocene–Holocene [17,18]).

Larvae of Diptera are notoriously underrepresented in the paleontological literature. It has been thought that this was due to preservation biases, but recent publications show that they are in fact abundant and have mostly been overlooked or ignored [19,20,21,22]. Associating fossilized immatures with their adult counterparts is extremely challenging, unequivocally possible only in exceptional cases, such as inclusions of larval exuviae attached to the pupa or a pharate adult entrapped during mid-emergence (examples in [9,23]). Even so, single immatures can still provide relevant insights into their habitats and shed light on morphological evolution, phylogenetic relationships, and especially ecological functions [8,24,25,26].

Amber deposits from Kachin, Myanmar, are approximately 100 Myo [27] and are an important source of information about the Cretaceous biota. This biota comprises a sample of tropical and subtropical species from the supercontinent Gondwana [28], which was formed by the extant South America, Africa, Madagascar, Arabia, India, Australia, Tasmania, New Guinea, New Zealand, New Caledonia, and Antarctica [29]. The Cretaceous saw remarkable radiations of many groups associated with a deep transformation of terrestrial ecosystems. Most of the major dipteran lineages were present by the end of the Cretaceous [2,30], coexisting with now extinct groups, which we still know little about.

The internal phylogeny of Diptera is a topic of much debate, with special lack of resolution in certain relationships, for instance, the early branchings of the group Brachycera (often unfortunately termed “lower Brachycera”). It is also important to view this phylogeny in the context of the paraphyletic nature of “Nematocera”, as Bibionomorpha are the likeliest sister group to Brachycera s.str [31].

Brachycera are a well-supported monophyletic lineage, represented by the more robust flies, with (often) sponge-like mouthpart apparatus and generally short antennae (sometimes collectively named “short-horned flies”). The early divergences within Brachycera include some historically recognized groups [32] such as Stratiomyomorpha (soldier flies and allies), Tabanomorpha (horse and snipe flies), and Xylophagidae (awl-flies; often put in the “empty bracket” Xylophagomorpha). These three groups are constantly being reinterpreted and rearranged in every new proposed phylogenetic reconstruction. For Xylophagidae, some phylogenies suggest a divergence within Brachycera as the sister lineage to all remaining brachyceran lineages [33,34,35,36]. Others have proposed a closer relationship of Xylophagidae with Tabanomorpha [37,38,39], or even with Stratiomyomorpha [40].

So far, fossil larvae of these three lineages have been relatively rarely recorded, besides the larvae of Stratiomyomorpha. These have been reported from various deposits [10] and appear to be dominating in the biota of the Kachin amber forest [19]. Few strongly derived larvae of Tabanomorpha have been reported from Jurassic deposits [41]. In the Cretaceous many further derived lineages of Brachycera were not yet present; the many ecological functions fulfilled by these, such as necrophagy, must have been performed by other animals, potentially also by larvae of the early branchings within Brachycera [19].

We here report new fossil larvae of these early branchings within Brachycera. With this, we expand the record of such larvae. We discuss the possible ecological roles of these new larvae and the impact of these new forms on our understanding of the early evolution of Brachycera.

## 2. Materials and Methods

### 2.1. Material

In total two slabs and 42 amber pieces were studied. Some amber pieces included more than 1 specimen resulting in a total of 55 studied specimens. Studied specimens come from two collections. One specimen (BUB 4258) comes from the collection of one of the authors (PM); the other specimens are deposited at the Palaeo-Evo-Devo Research Group Collection of Arthropods, Ludwig-Maximilians-Universität München, Germany, under the following repository numbers: PED 0841, 1184, 1435, 1606, 1767, 1832, 1834, 1858, 1861, 1879, 1880, 1892, 1893, 1919, 1938, 1943, 1963, 1964, 1965, 2000, 2012, 2157, 2159, 2233, 2257, 2279, 2287, 2347, 2450, 2505, 2562, 2595, 2651, 2659, 3127, 3176, 3225 3408, 3470, 3509, 3553, 3591, and 3650. All specimens with a PED label were legally purchased on the trading platform eBay from various traders (amber_stones, burmite-miner, collection-lovers, and meteorites-fossils).

Two specimens (PED 3176, 3225) originate from the Jurassic of Daohugou, China (see [41]). All other specimens are preserved in Kachin amber from Myanmar and date from the Cretaceous, Early Cenomanian, 98.17–99.41 Ma [27].

### 2.2. Documentation Methods

Most specimens were documented with optical microscopy as compound images. They were recorded using a Keyence VHX-6000 digital microscope (Keyence, Neu-Isenburg, Germany). Amber pieces were prepared with a drop of glycerine and a coverslip on top, to even the surface. Illumination was either unpolarized white ring light or cross-polarized coaxial light, depending on which retrieved better contrast and visibility. For each image, multiple stacks were recorded, which were then fused by the built-in software of the microscope. For one specimen, images from different angels of view were recorded. These half-images were combined into a red-cyan stereo image with Affinity Photo 2 (Affinity, v2.4).

Two of the amber pieces were scanned at the Imaging Beamline P05 [42,43], operated by the Helmholtz-Zentrum Hereon at the PETRA III storage ring (Deutsches Elektronen Synchrotron—DESY, Hamburg, Germany). The effective voxel size of the reconstructed volume is 2.56 µm. These micro-CT scans were then processed using FIJI/ImageJ, and the volume reconstructions were obtained using the open-source software 3D Slicer (v5.6.0).

### 2.3. Measurements

Specimens were measured from the optical microscopy images, using the open-source softwares GIMP (v2.10) and Fiji-ImageJ (v2.14).

## 3. Results

### 3.1. Short Descriptions of Specimens from Daohugou

PED 3176: Elongate larva, length 15.155 mm, preserved in dorsal view (Figure 1A). Head 0.222 mm long, barely distinguishable, without any distinguishable structure. Thorax length 2.031 mm. Individual segments not recognizable. Medial portion with impression of attachment structure composed of six concentric rims. Abdomen with seven segments, plus trunk end (possibly the fusion of multiple segments). Abdominal segments 1–7 length 9.422 mm; trunk end 0.951 mm long. Well-developed prolegs visible on segments 1–5, bearing distal crochets (about 8–10). Segments 6–7 with elongate lateral projections. Posterior end with pair of strong projections 2.528 mm long, bearing marginal fringes.

PED 3225: Elongate larva, length 20.635 mm, preserved in ventral view (Figure 1B,C). Head undistinguishable. Thorax length 2.906 mm. Individual segments not recognizable. Medial portion with ventral attachment structure composed of six concentric rims. Abdomen with seven segments, plus trunk end (possibly the fusion of multiple segments). Abdominal segments 1–7 length 12.773 mm; trunk end 1.311 mm long. Well-developed prolegs visible on segments 1–6, bearing two rows of distal crochets. Segments 6–7 with elongate lateral projections. Posterior end with pair of strong projections 3.644 mm long bearing marginal fringes.

### 3.2. Specimens of Stratiomyomorphan Morphotype 1 or Unclear Type

Most specimens represent stratiomyomorphan larvae of morphotype 1. These larvae present autapomorphies of Stratiomiomorpha, such as the distinct mandibular-maxillary complex and integument covered by pellets in a honeycomb pattern. This morphotype can be distinguished from the others by the slenderer body, presence of long triangular spines on the tergites, and smaller rounded spines on the sternites (detailed description in [10,19]). In total 26 specimens were examined: PED 1606 (Figure 2A), 1832 (Figure 2B,C), 1834 (Figure 2D,E), 1861 (Figure 3A, 5 specimens), 1880 (Figure 3E), 1892 (Figure 3C,D), 1893 (Figure 3B), 1938 (Figure 4A), 1943 (Figure 4D), 1963 (Figure 4C), 1965 (Figure 4B), 2233-1 (Figure 5A), 2233-2 (Figure 5B), 2450 (Figure 5C, 2 specimens), 2651 (Figure 5D), 2659 (Figure 6D), 3509-1 (Figure 6A), 3509-2 (Figure 6B), 3509-3 (Figure 6C), 3509-4 (Figure 6C), and 3650 (Figure 7F).

In addition, there are seven specimens not very well preserved that likely also represent stratiomyomorphan morphotype 1: PED 1951 (Figure 7A, 2 specimens), 2000, (Figure 7C), 2012 (Figure 7B), 2287 (Figure 7D), 2505 (Figure 7G), and 2562 (Figure 7E).

### 3.3. Specimens of Stratiomyomorphan Morphotype 2 or 7

Numerous specimens represent stratiomyomorphan larvae of morphotype 2 or 7. Apart from apomorphic Stratiomiomorpha traits (see Section 3.2 above), these specimens differ from morphotype 1 by the more robust body, head moderately to strongly protruded, lack of long dorsal spines (detailed description in [10,19]). Due to preservation conditions, it is not possible to place these specimens among morphotypes 2 or 7 with certainty. In total 11 specimens were examined: PED 1858 (Figure 8A), 1879 (Figure 8C), 1919 (Figure 8B), 1964 (Figure 8D), 2157 (Figure 9A), 2257 (Figure 9B), 2279 (Figure 9D), 2347 (Figure 9C), 2595 (Figure 10A), 3127 (Figure 10B), 3553 (Figure 10C).

### 3.4. Short Descriptions of Specimens of Elongate Chimera Morphotype

BUB 4258: Elongate larva, length 3.558 mm, accessible in lateral view (Figure 11A,B), other orientations virtually accessible (Figure 11C). Head more darkly pigmented, presumably more sclerotized than remaining body. Cuticle of trunk tessellated. Head 0.228 mm long. Head elongate and protruded anteriorly, with a ventral furrow, bearing vertically moving, strongly developed mouthparts (Figure 11E). Mandible and visible proximal part of maxilla robust and blade-like (Figure 11D). Distally, palpus seemingly with two elements. Head with one pair of dorsal setae at mid-length, single lateral seta at mid-length, and two pairs of ventral setae, one more anteriorly and other more posteriorly. Presence of lateral elevation on head capsule with thinner cuticle, possibly indicating stemma (Figure 11D). Internally, pair of slender tentorial rods present; further posteriorly, pair of slender metacephalic rods extending into prothorax (Figure 11F). Length of three thoracic segments combined 0.650 mm. Prothorax with lateral spiracle opening. Meso- and metathorax with dorsal projections, possibly a preservation artefact. Each thoracic segment at least with pair of ventral setae (Keilin’s organ). Abdomen with seven segments, plus trunk end (possibly compound of multiple segments). Abdominal segments 1–7 length 2.450 mm; trunk end 0.216 mm long. Abdominal segments 2–7 each with a pair of ventral prolegs, located anteriorly on the segment, each bearing a single elongate hook distally. Trunk end with single larger ventro-median compound proleg, seemingly lacking distal hook. Posterior end with pair of thin elongate projections, 0.166 mm long.

PED 1435: Elongate larva, length 4.329 mm, accessible in lateral view (Figure 12A,B). Head and trunk end more sclerotized than remaining body. Cuticle of trunk with distinct tightly spaced dorso-ventral folds (Figure 12C). Head 0.224 mm long, elongate and protruded anteriorly, with vertically moving mouthparts. Details of mouthpart structures not clearly visible. Presence of lateral elevation with thinner cuticle, possibly indicating stemma. Length of three thoracic segments combined 0.753 mm. Thorax without any visible appendices or projections. Abdomen with seven segments, plus trunk end (possibly compound of multiple segments). Abdominal segments 1–7 length 2.919 mm; trunk end 0.241 mm long. Segments 2–7 each with a pair of ventral prolegs, located anteriorly on segment, each bearing single elongate hook distally. Trunk end with single larger ventromedian compound proleg, presence of distal hook uncertain (Figure 12D). Posterior end with pair of thin elongate projections, 0.192 mm long.

PED 3591: Elongate larva, specimen strongly crumpled, length approximately 3.020 mm, accessible in lateral view (Figure 13A,B). Head more sclerotized than remaining body, about 0.240 mm long, protruded anteriorly, mouthparts not visible. Cuticle of trunk with distinct tightly spaced dorso-ventral folds. Thorax without visible appendices or projections. Abdomen with seven segments, plus trunk end (possibly compound of multiple segments). Abdominal segments 1–7 length about 2.230 mm; trunk end about 0.210 mm long. Abdominal segments 2–7 each with pair of ventral prolegs, located anteriorly on segment, each bearing single elongate hook distally. Trunk end with single larger ventromedian compound proleg, lacking hook. Posterior end with pair of thin elongate projections, about 0.150 mm long.

PED 3470: Elongate larva, length 5.167 mm, accessible in oblique dorso-lateral view. (Figure 14A,B). Cuticle of trunk with distinct tightly spaced dorso-ventral folds. Head more darkly pigmented, presumably more sclerotized than remaining body. Head 0.247 mm long, elongate and protruded anteriorly, bearing vertically moving, strong mouthparts. Detail of mouthparts and head structures not clearly visible. Lateral elevation of possible stemmata not distinguishable. Internal structures not visible. Length of three thoracic segments combined 1.029 mm. Spiracular opening on prothorax not visible. Meso- and metathorax without dorsal projections. Abdomen with seven segments, plus trunk end (possibly composed of multiple segments). Abdominal segments 1–7 length 3.273 mm; trunk end 0.342 mm long. Prolegs only visible on abdominal segments 2–3 due to position of specimen in amber. Posterior end with pair of thin elongate projections, 0.276 mm long.

PED 3408: Elongate larva, length 2.640 mm, accessible in dorsal view (Figure 14C). Specimen very damaged, no further detail visible. Indications of surface structure similar to that of other specimens.

PED 2159: Elongate larva, length about 4.731 mm, accessible in dorsal view (Figure 14D). Cuticle of trunk with distinct tightly spaced dorso-ventral folds (Figure 14E). Head missing. Length of three thoracic segments combined 0.899 mm. Spiracular opening on prothorax not visible. Meso- and metathorax without dorsal projections. Abdomen with seven segments, plus trunk end (possibly composed of multiple segments). Abdominal segments 1–7 length 3.495 mm; trunk end 0.337 mm long. Prolegs not visible due to position of specimen. Posterior end with a pair of thin elongate projections, 0.271 mm long.

PED 1767: Elongate larva, strongly wrinkled, length about 3.244 mm, accessible in oblique dorso-lateral view. (Figure 14F). Head not visible. Trunk segmentation not distinguishable. At least five abdominal segments with parapods. Posterior end apparently missing elongate projections.

### 3.5. Short Descriptions of Specimen of Stouter Chimera Morphotype

PED 1184: Robust larva, length 4.070 mm. (Figure 15A,B). Cuticle unpatterned. Head more darkly pigmented and presumably more sclerotized than remaining body. Head 0.290 mm long, elongate and protruded anteriorly. Mouthparts not visible. Setation not visible. Presence of lateral elevation with thinner cuticle, possibly indicating stemma. Tentorial rods not visible. Internally with pair of slender metacephalic rods extending into prothorax. Length of three thoracic segments combined 0.830 mm. Spiracular opening on prothorax uncertain. Meso- and metathorax without dorsal projections. Abdomen with seven segments, plus trunk end (possibly compound of multiple segments). Abdominal segments 1–7 length 3.273 mm; trunk end 0.300 mm long. Presence of prolegs uncertain. Posterior end with pair of thin elongate projections, 0.200 mm long.

### 3.6. Short Descriptions of Specimen of Crochet-Bearing Morphotype

PED 0841: Robust larva, with anterior portion severely wrinkled, total length 2.773 mm, accessible in ventral (Figure 16A) and dorsal view (Figure 16B). Head extremely elongate and slender, 0.374 mm long (Figure 16C). Mouthparts indistinguishable. Elements of internal head structure partially visible, composed of paired structure, supposedly tentorial rods. Length of three thoracic segments combined 0.840 mm. Spiracular opening on prothorax uncertain. Abdominal segments 1–7 length 1.318 mm; trunk end 0.241 mm long. Abdominal segments hard to distinguish but at least segments 1–6 each with pair of short prolegs bearing a ring of short crochets (Figure 16B,D–H). Posterior end apparently lacking pair of elongate projections.

## 4. Discussion

### 4.1. Additional Specimens of Known Morphotypes: Qiyia jurassica Chen et al., 2014

Previously described compression fossils from the same Lagerstätte as the specimens here reported, also strongly resembling these, have been formally described as *Qiyia jurassica* [41]. Among the distinct features are the distinctive arrangement of prolegs (one pair of crocheted prolegs on each of abdominal segments 1–6, plus a single medial proleg on segment 7), a robust thorax without visible segmentation, and the extremely conspicuous attachment structure (Figure 1), referred by Chen et al. [41] as a sucking disk. No other known species, fossil or extant, has such a structure (larvae of Deuterophlebiidae have similar structures, but differ in their morphology and arrangement; see Chen et al. [41] for further discussion); therefore, it is very likely that the two new fossil specimens are also representatives of *Q. jurassica*. Both specimens herein presented bear the same distinctive morphological features, although PED 3176 is preserved in dorsal view and PED 3225 in ventral view. A relationship (or an ingroup position) with the group Athericidae is highly likely, given that the combination of crocheted parapods on abdomininal segments with a fringed terminal process (as seen in the fossils) is a diagnostic feature of Athericidae [44,45].

### 4.2. Additional Specimens of Known Morphotypes: Stratiomyomorpha

The most common brachyceran larvae found in Myanmar amber are those of Stratiomyomorpha. We can report 44 additional specimens here. These strongly resemble larvae of stratiomyomorphan morphotypes 1, 2, and 7 (reported in earlier works; [10,19]) in overall habitus, but also in the details of the surface or spines.

### 4.3. Identity of New Morphotypes: Elongate Chimera Morphotype

Several of the herein-reported larvae from Myanmar amber (BUB 4258, PED 1435, 3591, 3470, 2159, and possibly also 3408; Figure 11, Figure 12, Figure 13 and Figure 14) appear to represent a single morphotype, the elongate chimera morphotype. Yet, specimen BUB 4258 differs from the others by the texture of the cuticle being tessellated (Figure 11A), while those of the others have distinct tightly spaced dorso-ventral folds (Figure 12C, Figure 13A and Figure 14A,E). Additionally, PED 1435 seems to have a more distinctly sclerotized trunk end. Both differences might be due to preservation conditions.

The combination of an elongate shape, lack of thoracic legs, and especially the presence of vertically moving mouthparts identify these larvae as representatives of the dipteran ingroup Brachycera. Within the group Brachycera, only some of the non-Muscomorphan groups, i.e., early-diverging brachyceran lineages, have larvae with externally protruded, variably sclerotized heads.

The combination of vertically moving mandibles (see above) with a strongly elongate, (somewhat) conical, and well sclerotized head in larvae has been considered a characteristic feature of Xylophagidae [33,36]. This character is very apparent in the elongate chimera morphotype. Additionally, the presence of a pair of long metacephalic rods extending into the prothorax in larvae (Figure 11F) is an autapomorphy of Xylophagidae (convergently present in Asilioidea and Empididae; [33]). An unusual feature of the new fossils, as lacking in all known extant larvae of Xylophagidae, is the presence of crocheted prolegs. Modern larvae bear at most a few sclerotized pads (creeping welts) on the ventral side of the abdomen [46].

Larvae of Athericidae (ingroup of Tabanomorpha) have a morphology and arrangement of abdominal prolegs similar to the larvae in amber. The particular arrangement of one pair of crocheted prolegs on each of the abdominal segments 1–7 plus a single ventromedian compound proleg on the trunk end has been considered unique to the group [44]. Yet, modern larvae of Athericidae differ from the fossils of the elongate chimera morphotype in the presence of a proleg on the first abdominal segment. Also, instead of having two rows of short crochets surrounding the proleg tip, seen in modern larvae, the fossils possess only a single elongated hook. Prominent crocheted prolegs on abdominal segments are also found on aquatic larvae of multiple distinct lineages of Diptera, such as Empididae (Clinocerinae and Hemerodromiinae), Ephydridae, Oreoleptidae (ingroup of Tabanomorpha), and the non-brachyceran groups Blephariceridae, Deuterophlebiidae, Nymphomyiidae, and *Dicranota* (ingroup of Limoniidae) [32,45,47,48,49]. Thus, the presence of prolegs in the larvae alone is not a strong signal for a relationship with Athericidae.

Another relevant feature is the distinctive trunk end (seemingly more sclerotised in PED 1435) with a pair of posterior filamentous projections. Larvae of Xylophagidae are characterised by a trunk end with sclerotised dorsal plates and a pair of hook-like processes [39] (p. 111). This indeed resembles the condition in some of the fossils (e.g., Figure 12D). Yet, in most specimens, the posterior processes do not really appear hook-like, but instead resemble the elongate processes of larvae of Athericidae, only lacking a fringe of fine setose projections, which is diagnostic for the group.

One more aspect arguing against interpreting the new larvae as representatives of Xylophagidae is the lack of adult counterparts. The oldest determined Xylophagidae fossils come from Eocene deposits [50]. Fossils of the groups *Ganeopteromyia* Mostovski, 1999, and *Sinonemestrius* Hong & Wang, 1990, dating from the Jurasssic and the Cretaceous, respectively, have initially been attributed to Xylophagidae, yet the systematic position of at least *Sinonemestrius* has been questioned; newer interpretations indicate an ingroup position within Heterostomidae (ingroup of Tabanomorpha; [51]). To date, no fossil larva of Xylophagidae has been reported.

The larvae of this morphotype, therefore, combine characters of Xylophagidae and Athericidae, resulting in a chimera-type morphology. This might mean that the larvae are representatives of one of the groups having evolved the characters of the other group convergently. Yet it can also not be easily excluded that Xylophagidae and Tabanomorpha are more closely related, and the combined characters of the fossil indicate a complex morphology of the larvae in the ground pattern of this group.

### 4.4. Identity of New Morphotypes: Stouter Chimera Morphotype

Specimen PED 1184 (Figure 15A) is largely similar in morphology to the elongate chimera type, yet with a much more robust body. Still, many of the characters also apply here, including for example the metacephalic rods (Figure 15A). It seems likely that it represents a closely related species. While many of the differences between the specimen of the elongate chimera morphotype can well be explained by preservation, we cannot exclude that it, in fact, represents several closely related species.

### 4.5. Identity of New Morphotypes: Crochet Bearing Morphotype

Specimen PED 0841 (Figure 16) differs from the other specimens especially in the morphology of the prolegs. They appear shorter and each bears a ring with numerous crochets. This aspect makes this fossil much more similar to modern larvae of Athericidae. Also, this larva has a well sclerotised elongate head capsule, resulting again in a chimera-type morphology.

### 4.6. Palaeoecology and Entrapment of Aquatic Organisms in Amber

A common trait to many larvae reported here (unclear for some) is the presence of crochet-bearing prolegs on many abdominal segments. This feature is almost exclusive to aquatic dipteran larvae [44,52]. This indicates that the fossil larva lived in aquatic environments. The extant larvae use the crochets to anchor themselves on the substrate or to grasp prey.

Although the biology of many remains largely unknown, the larvae of both Xylophagidae and Tabanomorpha are generally regarded as predators [46,53,54,55,56,57]. The mouthparts of one of the specimens, bearing sharp, blade-like mandibles and maxillae (Figure 11D), deeply resemble those of the predatory larvae of the extant representative of Xylophagidae *Exeretonevra angustifrons* Hardy, 1924 [55], also indicating a predatory behaviour for the fossils.

The preservation of aquatic organisms in amber is a topic of much debate. Baranov at al. [25] provided a thorough summary of the issue, highlighting that there are many ways in which aquatic organisms can get entrapped in tree resin, some of which could be involved in the preservation of specimens reported here. Amber-producing trees are common on the margins of aquatic environments [58], enabling the deposition of resin directly into water. Moreover, the variation of the water levels covering and exposing a resin flow may put aquatic organisms in contact with the resin [59]. When submerged in water, the resin takes longer to solidify, up to several weeks [60], which could allow the entrapment of organisms during this period. Another possible way, although less likely, is the fortuitous transport of specimens by wind from dried up water bodies to the tree trunks where resin is [61]. Overall, an interpretation of the fossils of the three chimera morphotypes (including the crochet-bearing one) as aquatic predators is well founded.

### 4.7. Seeming Rarity of Brachyceran Larvae in the Fossil Record

Immature stages of either fossil or extant dipterans are still largely understudied. This is an unfortunate fact, as it is in the larval stage that species usually perform most of their ecological interactions. This makes interpreting fossil brachyceran larvae even more challenging. It seems likely that this is the more important factor for the seeming rarity of fossil brachyceran larvae, rather than a true absence in the fossil record. The larvae reported here demonstrate that they are more common than often anticipated, as are unusual fossil forms, such as the larvae of *Qiyia jurassica*.

The new specimens of the stratiomyomorphan morphotypes underpin that they were dominating elements in the Cretaceous fauna [19] most likely fulfilling ecological functions occupied nowadays by larvae of more derived brachyceran groups, which had not yet evolved in the Cretaceous. The new morphotypes show that many ecological functions still remain to be discovered in Myanmar amber. These enigmatic larvae provide important indications on the ecology and phylogenetic relationships of early brachyceran flies, even though a precise systematic interpretation was not possible. With more studies on extant and extinct immature forms, we can have a better idea of the evolution of flies and understand the roles they played in palaeoecological networks.

## Figures and Tables

**Figure 1 insects-15-00270-f001:**
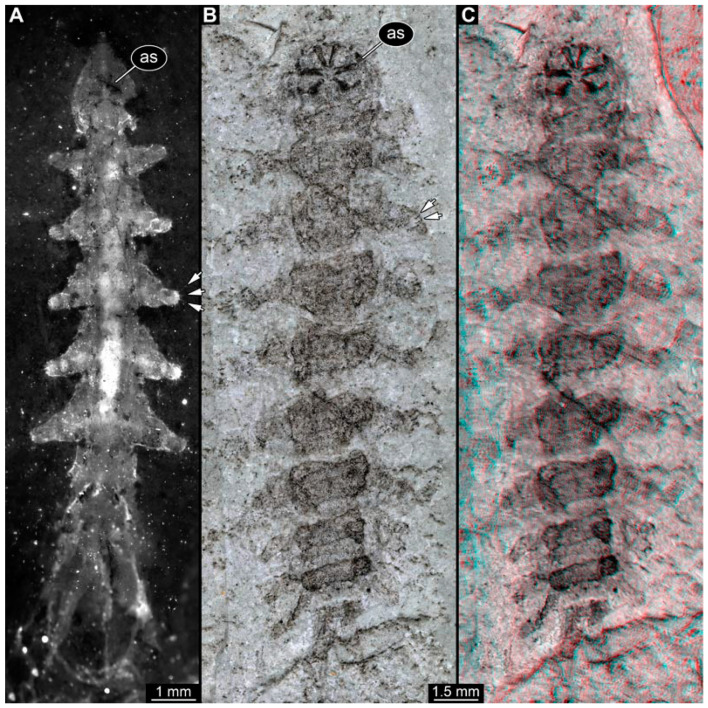
Additional larval specimens of *Qiyia jurassica* Chen et al., 2014, middle Jurassic, Daohugou, China. (**A**) PED 3176, dorsal view, composite autofluorescence image; arrows mark crochets on legs. (**B**,**C**) PED 3225, ventral view. (**B**) Composite image under cross polarised light; arrows mark crochets on legs. (**C**) Red-cyan stereo-anaglyph image highlighting the relief of the structurer; please use red-cyan glasses to view. Abbreviation: as = attachment structure formed by head and thorax region.

**Figure 2 insects-15-00270-f002:**
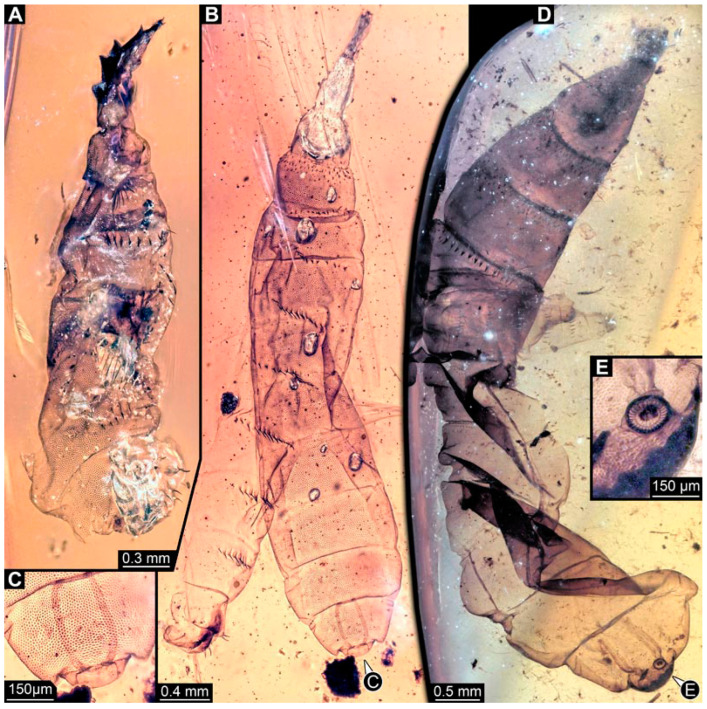
Larval specimens of stratiomyomorphan flies morphotype 1, late Cretaceous, Kachin amber, Myanmar. (**A**) PED 1606. (**B**,**C**) PED 1832. (**B**) Overview. (**C**) Close-up of posterior spiracles. (**D**,**E**) 1834. (**D**) Overview. (**E**) Close-up of posterior spiracle.

**Figure 3 insects-15-00270-f003:**
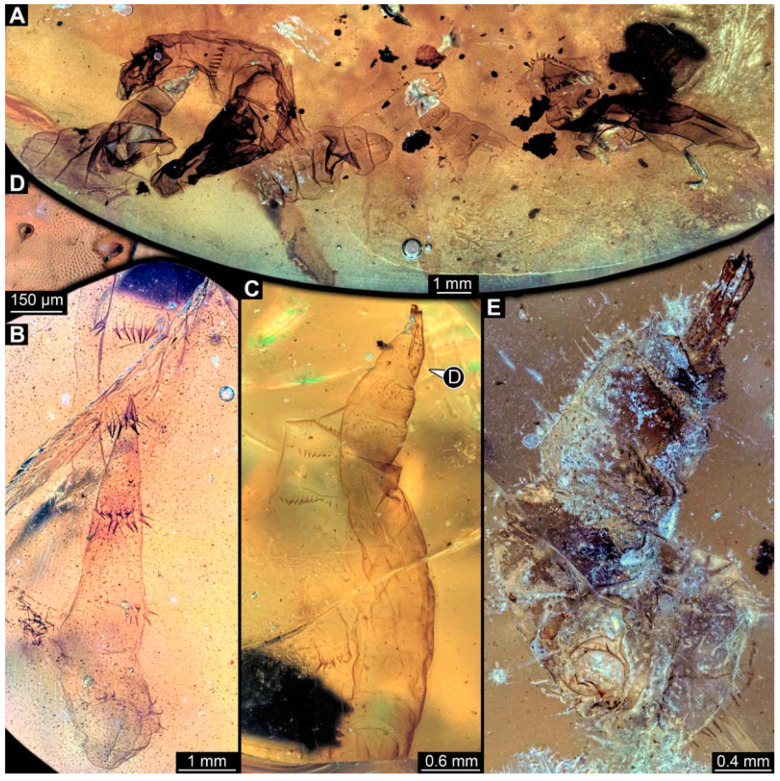
Larval specimens of stratiomyomorphan flies morphotype 1, late Cretaceous, Kachin amber, Myanmar, continued. (**A**) PED 1861 with remains of at least five larvae. (**B**) PED 1893. (**C**,**D**) PED 1892. (**C**) Overview. (**D**) Close-up of anterior spiracles. (**E**) PED 1880.

**Figure 4 insects-15-00270-f004:**
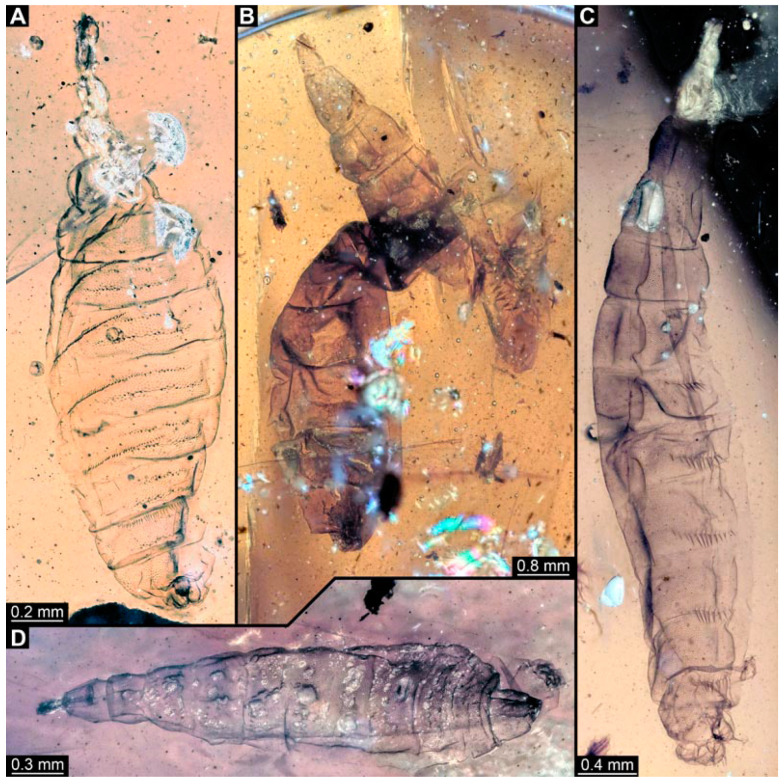
Larval specimens of stratiomyomorphan flies morphotype 1, late Cretaceous, Kachin amber, Myanmar, continued. (**A**) PED 1938. (**B**) PED 1965. (**C**) 1963. (**D**) 1943.

**Figure 5 insects-15-00270-f005:**
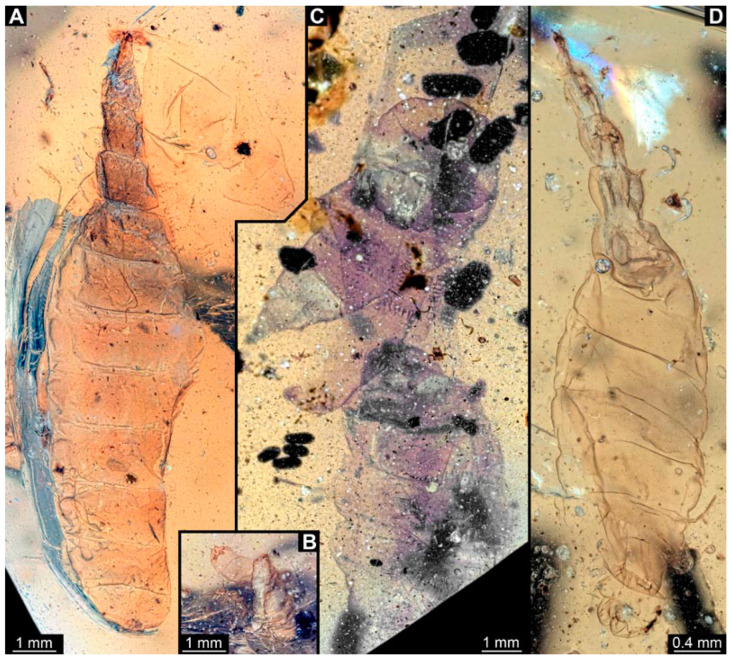
Larval specimens of stratiomyomorphan flies morphotype 1, late Cretaceous, Kachin amber, Myanmar, continued. (**A**) PED 2233-1. (**B**) PED 2233-2. (**C**) PED 2450, remains of two specimens. (**D**) PED 2651.

**Figure 6 insects-15-00270-f006:**
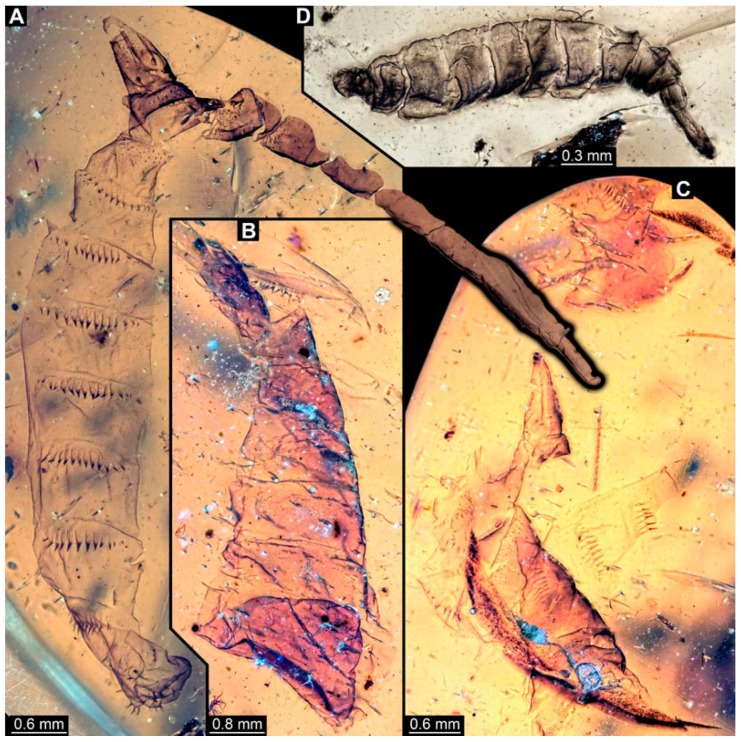
Larval specimens of stratiomyomorphan flies morphotype 1, late Cretaceous, Kachin amber, Myanmar, continued. (**A**) PED 3509-1. (**B**) PED 3509-2. (**C**) PED 3509-3 and 3509-4. (**D**) PED 2659.

**Figure 7 insects-15-00270-f007:**
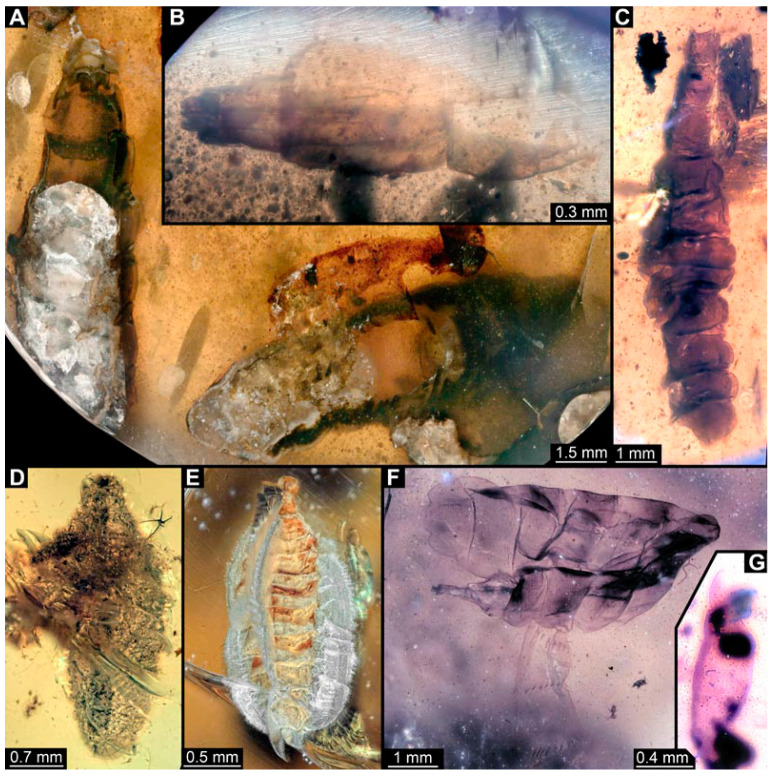
Larval specimens of stratiomyomorphan flies morphotype 1 and unclear specimens, late Cretaceous, Kachin amber, Myanmar. (**A**) PED 1951, with remains of two specimens. (**B**) PED 2012. (**C**) PED 2000. (**D**) PED 2287. (**E**) PED 2562. (**F**) PED 3650, morphotype 1. (**G**) PED 2505.

**Figure 8 insects-15-00270-f008:**
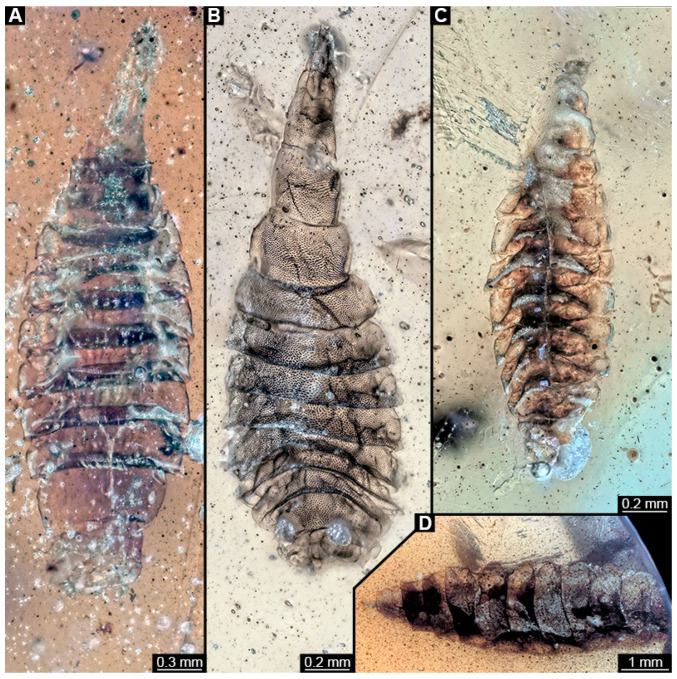
Larval specimens of stratiomyomorphan flies morphotype 2 or 7, late Cretaceous, Kachin amber, Myanmar. (**A**) PED 1858. (**B**) PED 1919. (**C**) PED 1879. (**D**) PED. 1964.

**Figure 9 insects-15-00270-f009:**
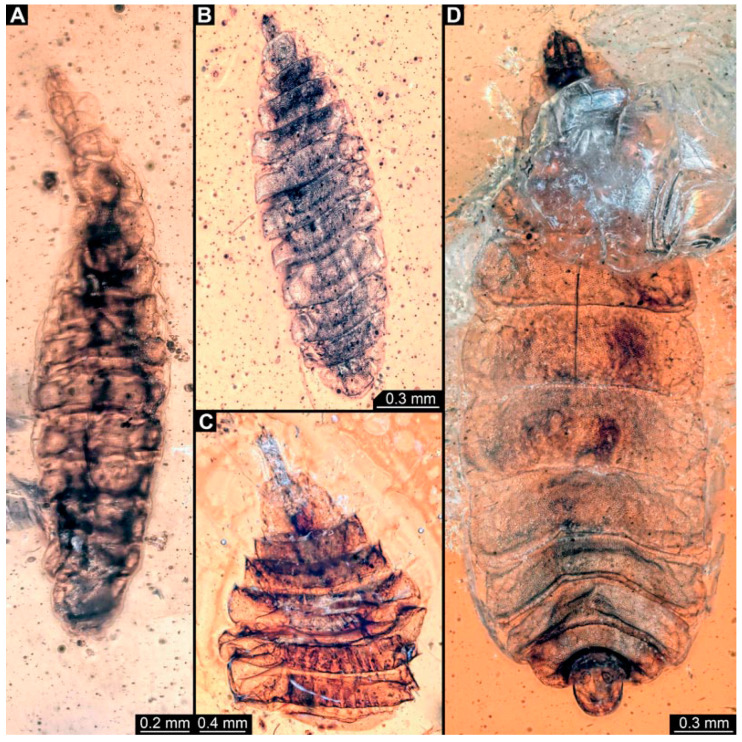
Larval specimens of stratiomyomorphan flies morphotype 2 or 7, late Cretaceous, Kachin amber, Myanmar. (**A**) PED 2157. (**B**) PED 2257. (**C**) PED 2347. (**D**) PED 2279.

**Figure 10 insects-15-00270-f010:**
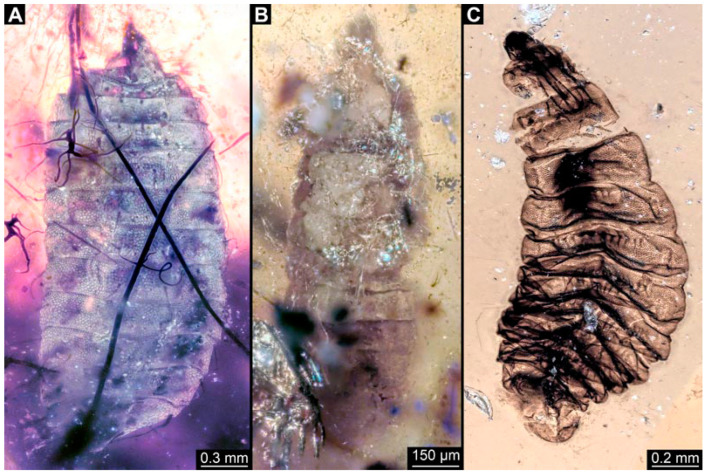
Larval specimens of stratiomyomorphan flies morphotype 2 or 7, late Cretaceous, Kachin amber, Myanmar. (**A**) PED 2595. (**B**) PED 3127. (**C**) PED 3553.

**Figure 11 insects-15-00270-f011:**
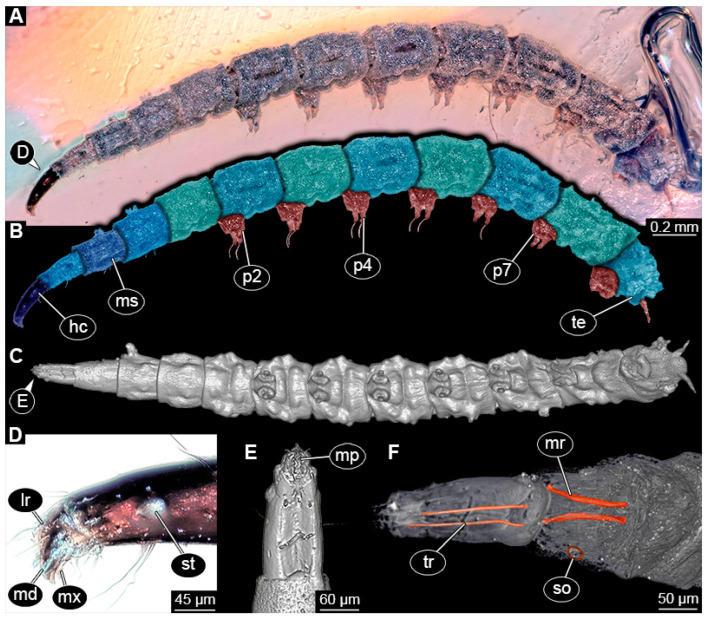
Specimen of new larval brachyceran elongate chimera morphotype, BUB 4258, late Cretaceous, Kachin amber, Myanmar. (**A**,**B**) Compound images, light microscopy. (**A**) Overview. (**B**) Colour-marked version of (**A**). (**C**) Overview in ventral view, surface model based on synchrotron-CT-scans. (**D**) Close-up on head region, compound image, light microscopy. (**E**) Close-up of head region, surface model based on synchrotron-CT-scans. (**F**) Close-up of anterior body region, volume rendering based on synchrotron-CT-scans. Abbreviations: hc = head capsule; lr = labrum; md = mandible; mp = mouthparts; mr = metacephalic rod; ms = mesothorax; mx = maxilla; p2–7 = proleg of abdominal segments 2–7; st = stemma; so = spiracle opening; te = trunk end; tr = tentorial rod.

**Figure 12 insects-15-00270-f012:**
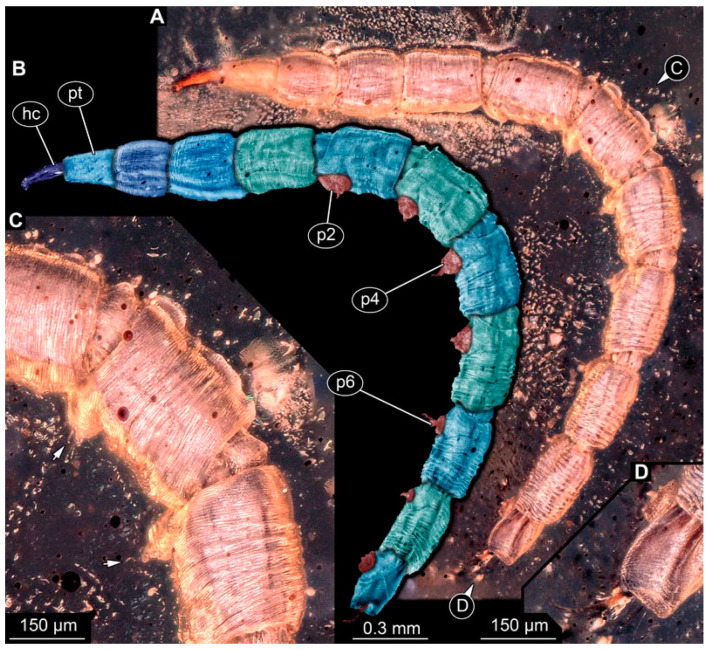
Specimen of new larval brachyceran elongate chimera morphotype, PED 1435, late Cretaceous, Kachin amber, Myanmar. (**A**) Overview. (**B**) Colour-marked version of (**A**). (**C**) Close-up of trunk surface with distinct folding pattern; arrows indicate hook at tip of proleg. (**D**) Close-up of trunk end. Abbreviations: hc = head capsule; p2–6 = proleg of abdominal segments 2–6; pt = prothorax.

**Figure 13 insects-15-00270-f013:**
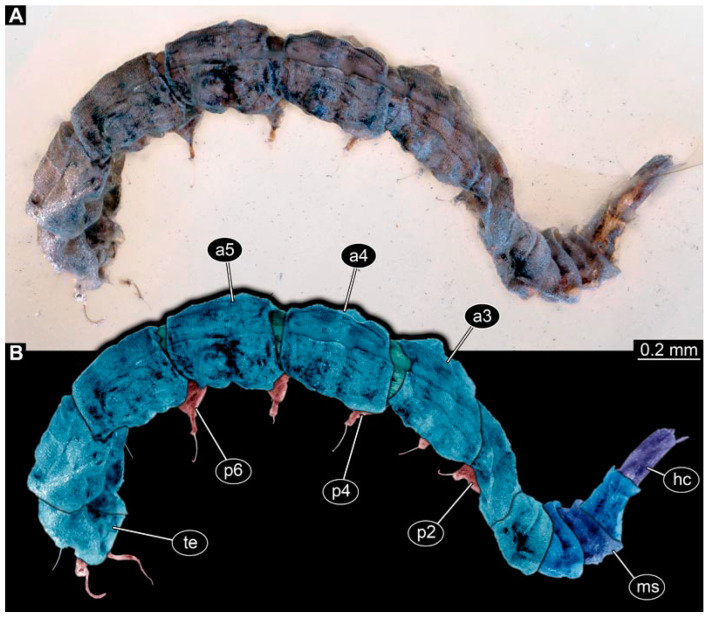
Specimen of new larval brachyceran elongate chimera morphotype, PED 3591, late Cretaceous, Kachin amber, Myanmar. (**A**) Overview. (**B**) Colour-marked version of (**A**). Abbreviations: a3–5 = abdominal segment 3–5; hc = head capsule; ms = mesothorax; p2–6 = proleg of abdominal segments 2–6; te = trunk end.

**Figure 14 insects-15-00270-f014:**
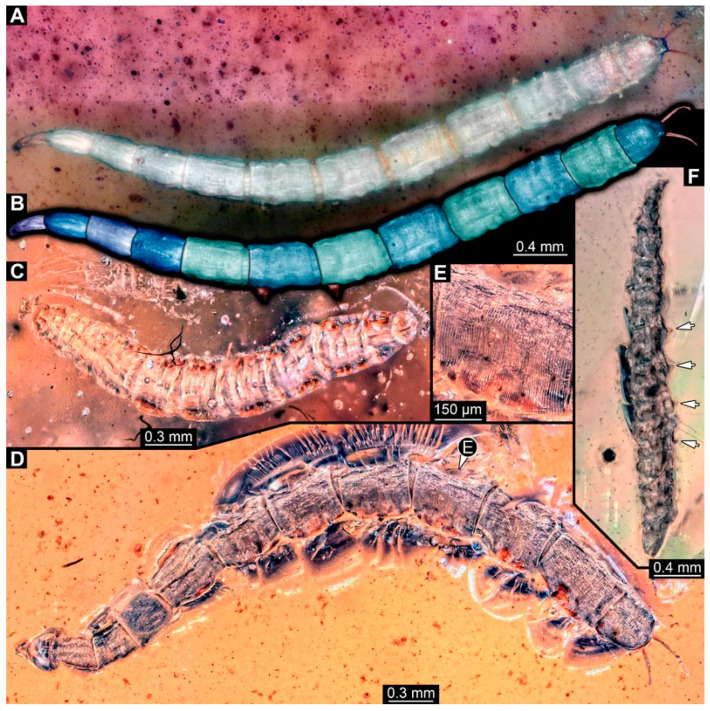
Specimens of new larval brachyceran chimera morphotypes, possibly all of the elongate type, late Cretaceous, Kachin amber, Myanmar. (**A**,**B**) PED 3470. (**A**) Overview. (**B**) Colour-marked version of (**A**). (**C**) PED 3408. (**D**,**E**) PED 2159. (**D**) Overview. (**E**) Close-up of trunk surface with distinct folding pattern. (**F**) PED 1767; arrows mark prolegs.

**Figure 15 insects-15-00270-f015:**
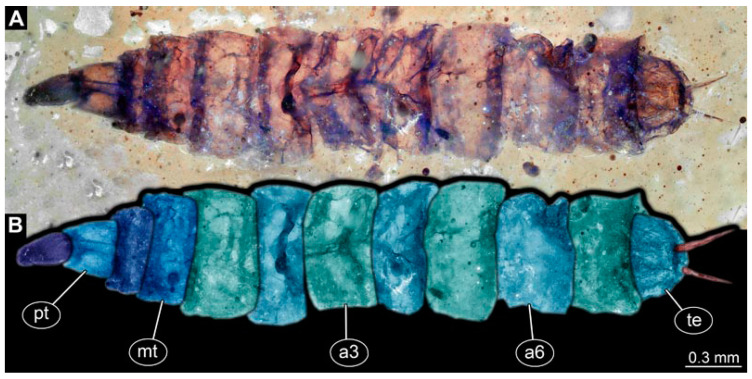
Specimen of new larval brachyceran stouter chimera morphotype, PED 1184, late Cretaceous, Kachin amber, Myanmar. (**A**) Overview. (**B**) Colour-marked version of (**A**). Abbreviations: a3, 6 = abdominal segments 3, 6; mt = metathorax; pt = prothorax; te = trunk end.

**Figure 16 insects-15-00270-f016:**
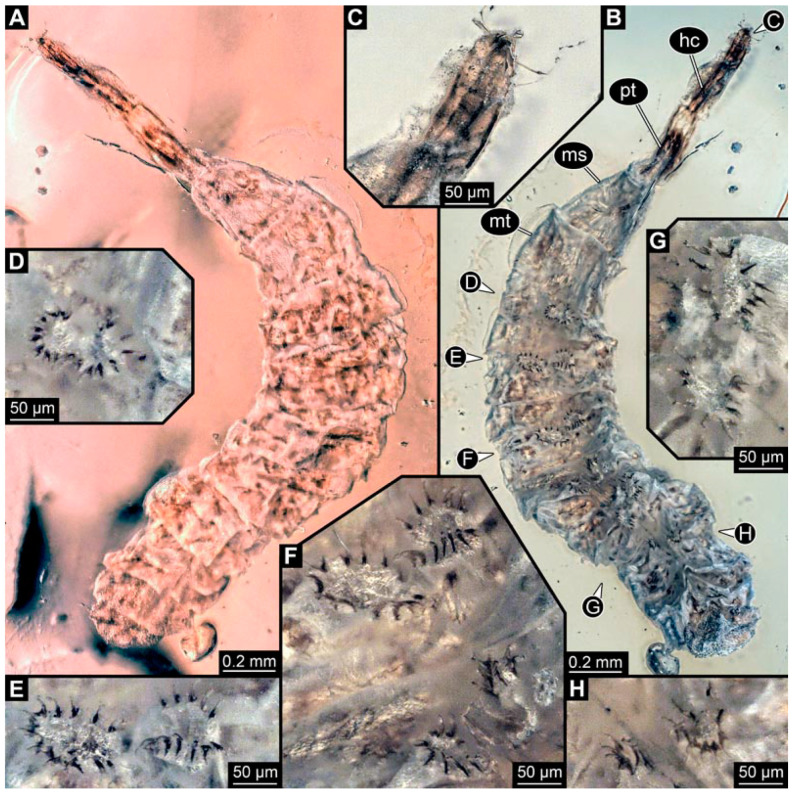
Specimen of new larval brachyceran crochet-bearing morphotype. PED 0841, late Cretaceous, Kachin amber, Myanmar. (**A**) Overview in dorsal view. (**B**) Overview in ventral view. (**C**) Close-up of anterior region of head. (**D**–**H**) Close-up of prolegs, with circles of crochets. (**D**) Abdominal segment 1. (**E**) Abdominal segment 2. (**F**) Abdominal segments 3 and 4. (**G**) Abdominal segment 5. (**H**) Abdominal segment 6.

## Data Availability

The raw data supporting the conclusions of this article will be made available by the authors on request.

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
