# Peer review of "Expanding the Mesozoic Record of Early Brachyceran Fly Larvae, including New Larval Forms with Chimera-Type Morphologies"

_insects, 2024, doi:10.3390/insects15040270_

Round 1
Reviewer 1 Report
Comments and Suggestions for Authors
This manuscript presents various primitive Brachyceran larvae based on fossils. Information on fossil larvae are rarely published, so this is a valuable contribution. The interpretations of the larval forms appear justify and many good points are raised. In addition to comments/ corrections written on the Word version of the manuscript, I have the following points:
1. The presentation of journal titles in the references is not always consistent. Sometimes written out in full or are abbreviated. Please check journal style.
2. Abdominal prolegs in the chimera larva is very interesting. Present day Xylophagids lack prolegs, but they live in rotting logs. Perhaps the evolution of prolegs is an adaptation to a more mobile lifestyle? The evolution of prolegs is certainly homoplastic and an adaptation to habitat. For example, Tabanidae larvae usually have only swellings and creeping welts, but in stream dwelling forms, prolegs have evolved (see Burger 1977, Trans Amer. Ent. Soc. 103: 145-258.) Prolegs are also known for early instar larvae of Bombyliidae - adaptation for seeking out hosts.
3. In Figs 2-6 the larvae appear to be "clear". Are these exuviae (note spelling)? ie., they do not appear to form puparia? Has this been discussed in other publications on fossil larval Stratiomyomorpha? In present day Stratiomyidae and Xylomyiidae, the final larvae instar forms a puparium.
4. The larvae of Fig. 8 do not appear to be exuviae. I can see facets on the cuticle of Fig. 8B. This is more typical of present day Strats.

Author Response
Comments included in the pdf file were replied in the pdf (attached).
- The presentation of journal titles in the references is not always consistent. Sometimes written out in full or are abbreviated. Please check journal style.
Fixed.
- Abdominal prolegs in the chimera larva is very interesting. Present day Xylophagids lack prolegs, but they live in rotting logs. Perhaps the evolution of prolegs is an adaptation to a more mobile lifestyle? The evolution of prolegs is certainly homoplastic and an adaptation to habitat. For example, Tabanidae larvae usually have only swellings and creeping welts, but in stream dwelling forms, prolegs have evolved (see Burger 1977, Trans Amer. Ent. Soc. 103: 145-258.) Prolegs are also known for early instar larvae of Bombyliidae - adaptation for seeking out hosts.
Considering that prolegs are usually present in predatory, free-living dipteran larvae, we can assume this is also true for our new specimens. Additionaly, the presence of crocheted prolegs (as far as we were able to verify) is only found in aquatic or semi-aquatic (such as some Eristalinae and Tabanidae) species.
- In Figs 2-6 the larvae appear to be "clear". Are these exuviae (note spelling)? ie., they do not appear to form puparia? Has this been discussed in other publications on fossil larval Stratiomyomorpha? In present day Stratiomyidae and Xylomyiidae, the final larvae instar forms a puparium.
- The larvae of Fig. 8 do not appear to be exuviae. I can see facets on the cuticle of Fig. 8B. This is more typical of present day Strats.
It is in general difficult to distinguish fossil fossil exuviae form carcasses (e.g. works by T.A. Hegna). Transparent appearing animals can look like that due to chemical processes during the time (or short) after embedding. A clearer sign would be ruptured moult sutures (as in some of the earlier reported stratiomyomorphans.

Reviewer 2 Report
Comments and Suggestions for Authors
General: The authors have done a good job at documenting the diversity of fossil immature flies, which was previously poorly documented, however for these larvae they have published most of the relevant information already. It is very difficult to link these fossil specimens to adult forms (not surprising since the immature stages are often poorly known even in extant Diptera), however the authors generally discuss these difficulties well. They are generally cautious about assigning the larval specimens to taxonomic groups, a caution I find to be justified.
My main concern is that the novelty of the present manuscript is quite low. Qiyia jurassica was described in greater detail in the original description by Chen et al (2014), I cannot see that the present manuscript adds any important information to this. As to the stratiomyiomorph larvae, they have been treated in detail in previous papers by several of the same authors. The morphotypes go back to Baranov et al. (2020) and a large number of specimens (110 of morphotype 1 alone) were treated by Amaral et al. (2023). This leaves the “chimera” larva as the only real novelty in this manuscript, and most of the manuscript does not really contain anything new of importance.
The treatment of Qiyia jurassica. I am a bit unsure why this is included in this paper: it is 50 million years older than the other taxa included in the paper, is a different type of fossil (compression fossils are often difficult to compare to amber fossils) and is obviously not closely related to the other types included. I have several questions concerning this larva.
The authors do not present strong arguments why (or if?) this larva should be placed in Brachycera. The fossils presented are short on anatomical detail, but from the general habitus it appears that they are most similar to the nematoceran families Deuterophlebiidae and, particularly, Blephariceridae, which are aquatic with large, crochet-bearing prolegs. Blephariceridae larvae also have suction pads apparently similar-ish to that seen in Qiyia, though the modern forms have several of them, not just one. I feel the authors should explain better why (if?) they consider Qiyia a brachyceran and not a blepharicerid-like nematoceran.
Figure 1, comparison of specimens PED 3176 and PED 3225: these two specimens the authors assign to the same species without much discussion. I find that their habitus looks fairly different both with regard to the shape of the head/head end with the attachment structure, and the hind end with abdominal processes. In principle, this could be due to the larvae being different instars since Diptera larvae often change considerably between different instars, however the size difference between the two specimens is not huge (c. 15 vs. 20 mm), accordingly, the authors should explain more carefully why they consider specimens 3176 and 3225 to belong to the same species, also perhaps add a better explanation as to why they consider this conspecific with (as opposed to just related to) the previously described Qiyia jurassica. Also in figure 1, I feel that Figure 1C does not add any more information compared to Figure 1B, and I feel it could be deleted without any loss of information.
The authors do well in documenting the various stratiomyiomorph larval types from amber, though I am not sure if their use of the term “chimera” is appropriate here. “Chimera” suggests a mixture of features belonging in different places, something abnormal. These larvae indeed have combinations of features not seen together in modern forms, but it occurs to me quite backwards to use the modern species as a “standard” of what is the “normal” distribution of features. There is no reason to think that there is anything abnormal about the morphology of these larvae, it is just different from the modern forms. This, of course, makes it harder to link them with adult forms, an issue I think the authors address well.
Line 100-101: would be nice to give some examples of which roles currently performed by derived Brachycera which may have been performed by others in the pasts, e.g. necrophagy?
Section 3.1 I suggest that, to simplify reading, you give measurements in mm and not μm here, 15 mm is more easy to comprehend than 15 000 μm. The same really applied thoughout, the authors generally do not refer to very minute structures here and mm are the appropriate unit of measurement in my view.
Figures 2 etc. I cannot see why it is essential to refer to all specimens as additional specimens in the figure legends. True, their classification refers to a previous paper by some of the same authors, they could still be referred to as just specimens. Specimens keep getting added all the time and that is how we (hopefully) gain better knowledge of the history of living things.
Line 375: the prolegs seen in larval Deuterophlebiidae and Blephariceridae, as discussed above, should probably also be mentioned here.
Literature
The literature list appears mostly sufficient, though some references to the larval morphology of Deuterophlebiidae and Blephariceridae (e.g., chapters from the various Diptera manuals) should be included.
Comments on the Quality of English Language
There are some minor language issues:
Lines 59-60: exceptions to exceptions? Suggest removing one of them
Line 60-61: larvae are extremely important in reconstructions of palaeoenvironments…
Line 83: do you mean that Bibionomorpha are likeliest to the sister group of Brachycera? A few words appear to be missing here.
Line 89: “empty bracket” is not a standard systematic term. Do you mean redundant group?
Line 91-92: I am not sure exactly what you mean here. Try and explain the relationship between Xylophagidae and other Brachycera better.
Line 95-96: except is probably better than besides here.
Line 99: suggests many of the derived lineages…
Line 123: optical microscope
Line 423: to grasp prey
Author Response
- My main concern is that the novelty of the present manuscript is quite low. Qiyia jurassica was described in greater detail in the original description by Chen et al (2014), I cannot see that the present manuscript adds any important information to this. As to the stratiomyiomorph larvae, they have been treated in detail in previous papers by several of the same authors. The morphotypes go back to Baranov et al. (2020) and a large number of specimens (110 of morphotype 1 alone) were treated by Amaral et al. (2023). This leaves the “chimera” larva as the only real novelty in this manuscript, and most of the manuscript does not really contain anything new of importance.
We underscore the importance of documenting newly discovered specimens. This is a way to show the different morphologies and their abundance relative to other forms. This data is important in reconstructing palaeoenvironments and understanding ecological roles in the past.
- The treatment of Qiyia jurassica. I am a bit unsure why this is included in this paper: it is 50 million years older than the other taxa included in the paper, is a different type of fossil (compression fossils are often difficult to compare to amber fossils) and is obviously not closely related to the other types included. I have several questions concerning this larva. The authors do not present strong arguments why (or if?) this larva should be placed in Brachycera. The fossils presented are short on anatomical detail, but from the general habitus it appears that they are most similar to the nematoceran families Deuterophlebiidae and, particularly, Blephariceridae, which are aquatic with large, crochet-bearing prolegs. Blephariceridae larvae also have suction pads apparently similar-ish to that seen in Qiyia, though the modern forms have several of them, not just one. I feel the authors should explain better why (if?) they consider Qiyia a brachyceran and not a blepharicerid-like nematoceran.
As the title indicates, our aim is to expand the Mesozoic record of early brachyceran fly larvae. Although we can be sure Qiyia jurassica is not conspecific of the other larvae presented, they might not be so distant phylogenetically. The “chimeric” larvae, as discussed, are possibly related to Xylophagidae or Athericidae (and these two groups may be close to each other). In the original paper describing Q. jurassica, the authors satisfactorily defended the position of this species as a representative of Athericidae lineage, so for this reason, we refrain from repeating the argument. In sum, the presence of paired crocheted parapods on all abdominal segments (plus a single proleg on segment 7), fringed terminal projections, and the reduced head and mouthparts (not visible in our specimens). The reduced head bearing mouth hooks (Chen et al. 2014, Fig. 1) is the strongest indication of the placement of this species among Brachycera. This species has such a peculiar morphology, notably, the enlarged undivided thorax bearing a ventral sucker, that there is little room for doubting that our specimens are either conspecifics or very closely related to each other.
Therefore, we consider that all specimens included in this manuscript represent Mesozoic lineages of brachyceran flies, which is the reason why they are together in this work.
- Figure 1, comparison of specimens PED 3176 and PED 3225: these two specimens the authors assign to the same species without much discussion. I find that their habitus looks fairly different both with regard to the shape of the head/head end with the attachment structure, and the hind end with abdominal processes. In principle, this could be due to the larvae being different instars since Diptera larvae often change considerably between different instars, however the size difference between the two specimens is not huge (c. 15 vs. 20 mm), accordingly, the authors should explain more carefully why they consider specimens 3176 and 3225 to belong to the same species, also perhaps add a better explanation as to why they consider this conspecific with (as opposed to just related to) the previously described Qiyia jurassica.
The main difference between the two specimens is that PED 3176 is in dorsal view and PED 3225 in ventral view (added to the caption). Upon closer inspection, we can identify the same structures in both specimens, although due to preservation artifacts they may seem distorted. We consider both conspecifics of Q. jurassica due to the unique morphology of these larvae, not remotely seen in any other known extinct or extant species, and they were all found in the same Lagerstätte. Of course, they might sister groups, but we can hardly assign multiple fossil specimens to the same species with so much certainty as in this case.
- Also in figure 1, I feel that Figure 1C does not add any more information compared to Figure 1B, and I feel it could be deleted without any loss of information.
If used with the appropriate glasses, the figure gives the idea of volume, which is an interesting information about the specimen and.
- The authors do well in documenting the various stratiomyiomorph larval types from amber, though I am not sure if their use of the term “chimera” is appropriate here. “Chimera” suggests a mixture of features belonging in different places, something abnormal. These larvae indeed have combinations of features not seen together in modern forms, but it occurs to me quite backwards to use the modern species as a “standard” of what is the “normal” distribution of features. There is no reason to think that there is anything abnormal about the morphology of these larvae, it is just different from the modern forms. This, of course, makes it harder to link them with adult forms, an issue I think the authors address well.
We agree with that. Please note that the term “chimera” is only applied to the specimens with traits common to Xylophagidae and Athericidae (Figs. 11–15).
- Line 100-101: would be nice to give some examples of which roles currently performed by derived Brachycera which may have been performed by others in the pasts, e.g. necrophagy?
Yes. Added to the text. Further discussion in Amaral et al. 2023.
- Section 3.1 I suggest that, to simplify reading, you give measurements in mm and not μm here, 15 mm is more easy to comprehend than 15 000 μm. The same really applied throughout, the authors generally do not refer to very minute structures here and mm are the appropriate unit of measurement in my view.
Fixed.
- Figures 2 etc. I cannot see why it is essential to refer to all specimens as additional specimens in the figure legends. True, their classification refers to a previous paper by some of the same authors, they could still be referred to as just specimens. Specimens keep getting added all the time and that is how we (hopefully) gain better knowledge of the history of living things.
Solved. Only kept in Figure 1 for Q. jurassica, to highlight these are not the type specimens.
- Line 375: the prolegs seen in larval Deuterophlebiidae and Blephariceridae, as discussed above, should probably also be mentioned here.
Added.
- Literature. The literature list appears mostly sufficient, though some references to the larval morphology of Deuterophlebiidae and Blephariceridae (e.g., chapters from the various Diptera manuals) should be included.
The association of Q. jurassica with Blephariceridae has been discussed by Chen et al. 2014. They convincingly argue that their species differs from blepharicerids in the structure of the sucking disk (striated in the fossil species and round in blepharicerids), besides the arrangement, of a single one near the head in Q. jurassica, versus a row along the body. Considering also that the Daohugou beds were low-energy lacustrine deposits, the habits of Q. jurassica were probably also different, i.e. not adapted to fast flowing streams. Finally, in their figure 1, it is possible to see the mouthparts typical of brachyceran flies. Regarding Deuterophlebiidae, these have suction disks at the apex of their paired prolegs, a distinctively different configuration, which gives no indication of homology.
Round 2
Reviewer 2 Report
Comments and Suggestions for Authors
General statement: This research group has published extensively on fossil larvae, which has been to some extent a neglected field. A consequence of that is that a substantial number of the works they cite are from their own research group. In my judgment this is not excessive self-citation, though, these works are the most relevant works published in this field in recent years, and provide a necessary background for interpreting the results. It does, however, raise some questions about the degree of novelty of the present manuscript, all except the “chimera” and “crochet”-larvae has essentially been published before and I find it difficult to see that the present paper adds anything relevant except for listing and illustrating more specimens from the same sources (mainly Myanmar amber).
This is particularly so for the treatment of Qiyia jurassica. The specimens treated here are from the type locality of the species, so they do not add any new distribution information. However, they are considerably less well preserved than the type material described in great detail by Chen et al. (2014), so they do not add any relevant information. Publishing accounts of additional specimens of already-described species from already-known localities will be relevant if these specimens add new information not available from previous specimens, but much less so if they don’t.
The authors have chosen to describe each specimen of the new morphotypes separately. This is a bit unconventional in systematic works, but probably justified here since, although these larvae are grouped together in morphotypes, it is highly probable that they do not all belong to one species.
Minor points: Author list: who is Jörg? I suppose everyone should be listed by full name.
Author Response
Concerning the idea of the reviewer to remove parts of the specimen from the manuscript: The suggestion of the reviewer is in line of the common habit to only report what is “new”. This is especially problematic when dealing with fossils as it leads to very awkward situations, and is even more expressed when it comes to fossil larvae. Until reports from our work group about brachyceran larvae in amber (e.g. Baranov et al. 2020 PeerJ, 2021 Palaeo-Electronica) these larvae were considered rare. Another report (Amaral et al. 2023 published in MDPI Insects) showed that such larvae are not only not rare, but in fact common and an important part of the fossil faunas. The report here indeed shows new forms of early branches in Brachycera, but in addition expands the record of already known forms (Stratiomymorpha and Athericidae). This emphasises how common and in part dominating these forms are and thus contributes important clues for reconstructing aspects of the fossil faunas.
Reporting only new forms has, for example, led to the awkward situation that fossil larvae of beetles are drastically underrepresented in the literature while they are in fact also quite common. We therefore can only emphasise how important it is, not only for the story of the manuscript but also in general, to not only report new forms, but additional finds of already known ones. We therefore think that these specimens need to remain as part of the manuscript.